# Understanding spatial effects in species distribution models

**Iosu Paradinas**[1,2]*, **Janine Illian**[2], **Sophie Smout**[2,3]

**1** Scottish Ocean's Institute, University of St Andrews, East sands, St Andrews, United Kingdom, **2** AZTI, Txatxarramendi Ugartea z/g, Sukarrieta, Bizkaia, Spain, **3** School of Mathematics and Statistics, University of Glasgow, Glasgow, United Kingdom

* ip30@st-andrews.ac.uk

## Abstract

Species Distribution Models often include spatial effects which may improve prediction at unsampled locations and reduce Type I errors when identifying environmental drivers. In some cases ecologists try to ecologically interpret the spatial patterns displayed by the spatial effect. However, spatial autocorrelation may be driven by many different unaccounted drivers, which complicates the ecological interpretation of fitted spatial effects. This study aims to provide a practical demonstration that spatial effects are able to smooth the effect of multiple unaccounted drivers. To do so we use a simulation study that fit model-based spatial models using both geostatistics and 2D smoothing splines. Results show that fitted spatial effects resemble the sum of the unaccounted covariate surface(s) in each model.

## Introduction

Understanding and predicting species spatial patterns through Species Distribution Models (SDM) is pivotal for ecology, evolution and conservation [1]. SDMs quantify the relationship between species occurrence or abundance with biotic and abiotic factors in order to gain ecological and evolutionary understanding [2, 3]. This way SDMs allow us to predict distributions across landscapes and make future predictions based on identified drivers. Ideally, these factors would fully explain the distribution of the species under study, but this is practically unfeasible due to the large number of factors that drive the distribution of a species. In this regard, by including a spatial effect into a SDM, one can accommodate the spatial structure of the data that is unaccounted by our covariates.

Spatial autocorrelation refers to the dependence between pairs of observations in space. In SDMs, spatial effects allow us to predict better and reduce Type I errors in the presence of covariates [4, 5]. In species distribution, spatial autocorrelation arise due to unaccounted environmental or biotic drivers that are often hard to measure or estimate, for example a geographical range dispersion process or a highly dynamic processes such as wind and current [6–8]. These drivers can influence species distribution at all scales, from micrometres to continental and ocean-wide scales [9]. However, the size, spacing and extent of sampling units will constrain the scale of inferable drivers, and the scale of spatial autocorrelation [8, 10]. In other words, if we sample at a kilometer scale, we cannot infer processes at a smaller scale, and

**Data Availability Statement:** This paper is based on a simulation study. All R code is available in https://github.com/iparperspective/Understanding-spatial-effects-.

**Funding:** IP was funded by a Marie Skłodowska-Curie Research Fellowship (GAP-847014). The funders had no role in study design, data collection and analysis, decision to publish, or preparation of the manuscript.

**Competing interests:** The authors have declared that no competing interests exist.

inversely, if our study area is one kilometer long, we cannot infer processes that affect at a larger scale.

The statistical interpretation of a spatial effect is related to the sign and link function of our linear predictor, but in general terms, positive values refer to areas where we expect more than that predicted by the rest of the linear predictor and vice versa. Ecologically, many SDM studies have linked spatial effects to biological features like home-range [6], hot-spot size [11] and unaccounted environmental drivers [12], providing reasonable arguments. For example, given a species that is driven by two environmental variables, one that drives the large-scale variation and another that drives the small-scale variation, the residual spatial pattern of a SDM that includes one of the two covariates will resemble the pattern of the unaccounted explanatory variable, either the large-scale or small-scale one. However, as we mentioned before, reality behind ecological processes is often high dimensional and variables that drive spatial correlation can occur at several different scales. In fact, SDMs are seldom able to identify more than a small portion of all the drivers that influence the distribution of the species under study. This results on spatial effects that are potentially driven by many different unaccounted drivers, diluting their interpretability in terms of an individual process. Although this interpretation issues have sporadically been addressed in the literature [7, 8, 13–17], many modellers fail to acknowledge this probably due to the lack of an explicit study that shows this.

The aim of this article was to complement the MBGapp Shiny App, a user-friendly interface created by [18], used for teaching geostatistical analysis to scientists with only minimal statistical training. We provide a practical demonstration that spatial effects are able to smooth the effect of multiple unaccounted drivers, making the biological interpretation of spatial effects complicated. To do so, we used spatial models applied over simulated species distribution surfaces. Simulated fields were based on three spatially structured environmental covariates acting at different spatial scales, and a geographical range dispersion process.

## Methods

We used an iterative simulation approach to produce spatially aggregated distributions that include a dispersion effect. At each iteration we added a fixed number of new specimens to the study area based on a probability surface constituted by three spatially structured covariates, each operating at different scales (i.e., small, medium and large scale), and a spatial dispersion process driven by the abundance of the neighbouring areas, mimicking the colonization of a plant species for example. As a result, our simulated species abundace were driven by the sum of four different effects (Fig 1): the influence of three explanatory environmental variables operating at different spatial scales (S = small, M = medium and L = large) and a spatial dispersal effect that increase the spatial autocorrelation of the response variable.

We simulated fifty different scenarios, selected 100 random samples for each scenario and fitted all the possible combinations of Poisson spatial models that ranged from a purely spatial model to a full model that accounted for the three covariates (see Table 1). We used two spatial modelling approaches, geostatistics through the Integrated Nested Laplace Approximation approach (INLA) [19] and 2D smoothing splines through the MGCV package for R [20, 21] using their defaults settings.

Our aim was to assess the resemblance between fitted spatial effects and unaccounted covariate surface combinations. Resemblence was assessed through the similarity in pattern score (SIP) [22]. SIP scores are bound between zero and one, and high scores denote high similarity in pattern and vice versa. For each simulated scenario, we calculated the SIP score between the spatial effect of every fitted model (rows in Table 2) and all the possible different combinations of covariate surfaces (columns in Table 2), and recorded the absolute difference

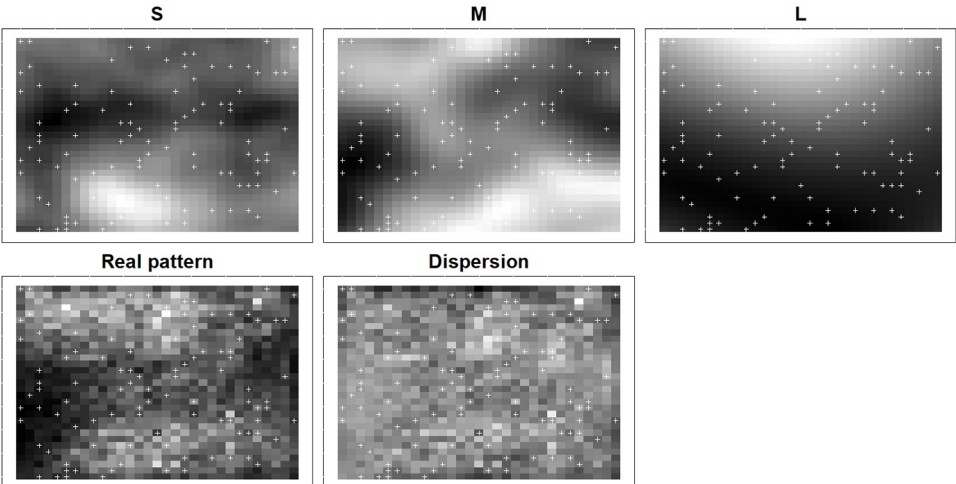

**Fig 1. Visualization of the different autocorrelated drivers that influence the abundance pattern in a simulated scenario.** S, M and L refer to the small, medium and large scaled covariate fields, respectively. Dispersion refers to the geographical range dispersion. White crosses refer to the simulated 100 samples.

between the best SIP score and the rest (i.e., SIP differences calculated per row in Table 2). This way, the spatial effect that best resembled a given combination of covariate surfaces scored a zero and that with the worst resemblance recorded the highest value (see Supporting information for a more detailed explanation of the procedure). As a result, we obtained fifty scores per model and combination of covariate surfaces. Finally, we summarised these scores by their mean and standard deviation. All the R script is available at https://tinyurl.com/2p8n3e4r.

## Results

Results show that fitted spatial effects resemble the sum of the unaccounted covariate surfaces in each model (see highlighted diagonal scores in Table 2). Fitted 2D splines using generalized additive models (GAM) seemed to perform a little worse than model based-geostatistics, probably due to the default selection of knots, but the overall pattern is very similar. This result suggests that spatial effects are able to smooth complex residual spatial patterns originated by a set of covariates that operate at very different scales. For example, model M_M, which only accounts for the mid-scale covariate, estimates a spatial effect that resembles the sum of the

**Table 1. Summary of fitted models.** W refers to a geostatistical spatial correlation term, S, M and L refer to the small, medium and large scale covariates, respectively.

| Model | Linear predictor | Missing covariates |
|---|---|---|
| M_0 | $\beta_0 + W$ | S, M & L |
| M_S | $\beta_0 + S + W$ | M & L |
| M_M | $\beta_0 + M + W$ | S & L |
| M_L | $\beta_0 + L + W$ | S & M |
| M_ML | $\beta_0 + M + L + W$ | S |
| M_SM | $\beta_0 + S + M + W$ | L |
| M_SL | $\beta_0 + S + L + W$ | M |
| M_SML | $\beta_0 + S + M + L + W$ | – |

**Table 2. Resemblance between fitted spatial effects, using geostatistics and 2D smoothing splines, against all the possible combinations of covariate surfaces (per simulation).** Scores must be read by row, and reflect the difference between the best SIP score and all possible combinations of drivers for each simulation and model. Therefore, lower values represent higher resemblance and have been highlighted in bold. We present the mean difference and standard deviation (in parenthesis). See Supporting information for a more detailed explanation of the procedure that we followed.

| | Model | Combination of drivers | | | | | | | |
|---|---|---|---|---|---|---|---|---|---|
| | | Dispersion | S | M | L | S & M | S & L | M & L | S, M & L |
| Geostatistics (INLA) | M_0 | 0.62 (0.14) | 0.30 (0.13) | 0.27 (0.18) | 0.35 (0.22) | 0.11 (0.06) | 0.17 (0.08) | 0.12 (0.15) | **0.01 (0.02)** |
| | M_S | 0.56 (0.18) | 0.66 (0.22) | 0.19 (0.16) | 0.25 (0.19) | 0.33 (0.17) | 0.41 (0.16) | **0.01 (0.03)** | 0.16 (0.12) |
| | M_M | 0.47 (0.17) | 0.19 (0.15) | 0.71 (0.25) | 0.26 (0.22) | 0.26 (0.21) | **0.04 (0.08)** | 0.37 (0.21) | 0.11 (0.14) |
| | M_L | 0.55 (0.17) | 0.21 (0.14) | 0.24 (0.21) | 0.78 (0.35) | **0.04 (0.04)** | 0.29 (0.20) | 0.33 (0.24) | 0.08 (0.12) |
| | M_SM | 0.34 (0.17) | 0.50 (0.24) | 0.61 (0.28) | **0.07 (0.13)** | 0.48 (0.26) | 0.22 (0.15) | 0.21 (0.18) | 0.24 (0.20) |
| | M_SL | 0.41 (0.23) | 0.51 (0.21) | **0.08 (0.11)** | 0.67 (0.35) | 0.18 (0.11) | 0.53 (0.25) | 0.17 (0.20) | 0.20 (0.16) |
| | M_ML | 0.36 (0.18) | **0.06 (0.10)** | 0.59 (0.24) | 0.65 (0.26) | 0.11 (0.11) | 0.13 (0.14) | 0.60 (0.24) | 0.16 (0.17) |
| | M_SML | **0.09 (0.15)** | 0.27 (0.16) | 0.40 (0.22) | 0.43 (0.24) | 0.25 (0.16) | 0.28 (0.18) | 0.40 (0.24) | 0.25 (0.22) |
| 2D splines (GAM) | M_0 | 0.50 (0.09) | 0.18 (0.10) | 0.11 (0.08) | 0.07 (0.07) | 0.09 (0.08) | 0.12 (0.08) | **0.04 (0.05)** | **0.04 (0.05)** |
| | M_S | 0.50 (0.09) | 0.38 (0.17) | 0.10 (0.10) | 0.08 (0.09) | 0.22 (0.14) | 0.28 (0.17) | **0.02 (0.04)** | 0.14 (0.11) |
| | M_M | 0.48 (0.10) | 0.15 (0.11) | 0.35 (0.19) | **0.03 (0.04)** | 0.24 (0.16) | **0.07 (0.08)** | 0.23 (0.17) | 0.15 (0.14) |
| | M_L | 0.33 (0.19) | 0.13 (0.13) | **0.06 (0.08)** | 0.16 (0.19) | **0.08 (0.10)** | 0.16 (0.17) | 0.11 (0.12) | 0.11 (0.13) |
| | M_SM | 0.49 (0.10) | 0.38 (0.17) | 0.36 (0.19) | **0.00 (0.02)** | 0.42 (0.22) | 0.23 (0.14) | 0.18 (0.14) | 0.28 (0.19) |
| | M_SL | 0.35 (0.17) | 0.25 (0.19) | **0.05 (0.08)** | 0.16 (0.17) | 0.16 (0.13) | 0.26 (0.19) | 0.10 (0.12) | 0.18 (0.13) |
| | M_ML | 0.33 (0.16) | **0.09 (0.10)** | 0.20 (0.19) | 0.14 (0.16) | 0.16 (0.14) | 0.13 (0.14) | 0.23 (0.19) | 0.18 (0.14) |
| | M_SML | 0.34 (0.17) | 0.23 (0.21) | 0.20 (0.20) | 0.14 (0.20) | 0.27 (0.20) | 0.26 (0.19) | 0.22 (0.18) | 0.28 (0.17) |

small-scale and large-scale covariate effects (S and L respectively). Similarly, the spatial effect of model M_0, which is a purely spatial model (no covariates included), mirrors the combination of all three covariate surfaces (S, M and L). In the particular cases where we included two covariates (i.e., only one unaccounted covariate), spatial effects resembled the missing covariate.

## Discussion

Many studies have analysed the characteristics of spatial effects to describe the unaccounted ecological mechanisms that drive the distribution of species and try to associate spatial effect patterns to single unaccounted drivers such as home range or dispersion [4, 14, 23]. However, most species distributions are driven by a large number of factors and we are seldom able to identify most of these drivers in our statistical models. As a consequence, SDM spatial effects constitute a combination of many unaccounted factors [6–8].

This study used a simulation study to illustrate the difficulty in interpreting spatial effects with regards to unaccounted environmental drivers. Here, we did not attempt to account for all possible cases, instead, we aimed to illustrate our point using a simple and intuitive approach. Fitted spatial effects resembled the sum of the unaccounted covariate surfaces, including spatial patterns originated by covariates that operated at very different scales. Therefore the biological interpretation of spatial effects may only be valid when the unexplained spatial heterogeneity of the data is characterised by a single dominant driver. At this point, the question is: how many times do SDMs account for all but one driver? One can only speculate this answer but our guess would be: hardly ever. The environmental and ecological processes that drive the distribution of species are complex and diverse, and one could only arbitrarily assume that there is only one covariate missing in our SDM predictor to make biological interpretations over fitted spatial effects.

In this regard, one could use a multiresolution decomposition approach to identify dominant features within the residual spatial correlation of the data [16, 17]. This method essentially estimates the range of spatial correlation at different resolutions of the data, or in this case, residuals of the SDM to help us identify the scale-dependent features within the spatial effect of the residuals. Then, assuming that each scale is characterised by a single dominant driver [13], one could relate them to underlying process generating mechanisms.

## Conclusions

Spatial autocorrelation is a common feature in ecological data. As a consequence, spatial correlation models are important to correctly estimate covariate standard errors and therefore reduce Type I errors. Additionally, spatial correlation terms estimate the residual spatial structure of the data, improving the predictive capacity of our models at locations that are within the sampled area. In ecology, residual spatial patterns are potentially driven by complex multivariate and multi-scaled systems, which can be accommodated by a single spatial effect. Therefore, the biological interpretation of spatial effects is very difficult. A multiresolution decomposition of residual spatial patterns [17] could help us identify the scale-dependent features within the spatial correlation structure of the residuals assuming that each scale is characterised by a single dominant driver.

## Supporting information

**S1 File.**
(ZIP)

## Author Contributions

**Conceptualization:** Iosu Paradinas, Janine Illian, Sophie Smout.

**Formal analysis:** Iosu Paradinas.

**Funding acquisition:** Iosu Paradinas, Janine Illian, Sophie Smout.

**Investigation:** Iosu Paradinas.

**Methodology:** Iosu Paradinas, Janine Illian, Sophie Smout.

**Writing – original draft:** Iosu Paradinas.

**Writing – review & editing:** Sophie Smout.

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
