## [Decision Letter · Decision Letter 0]

4 Oct 2022

PONE-D-22-21589Understanding spatial effects in species distribution modelsPLOS ONE

Dear Dr. Paradinas,

Thank you for submitting your manuscript to PLOS ONE. After careful consideration, we feel that it has merit but has the potential to be improved by following the reviewer's suggestions in the attached file

We look forward to receiving your revised manuscript.

Kind regards,

Judi Hewitt

Academic Editor

PLOS ONE

Journal Requirements:

 “IP was funded by a Marie Curie Research Fellowship (GAP-847014)”     

“IP is grateful to the MSCA fellowship that supported your research.”

“IP was funded by a Marie Curie Research Fellowship (GAP-847014)”      

Reviewers' comments:

Reviewer's Responses to Questions

**Comments to the Author**

1. Is the manuscript technically sound, and do the data support the conclusions?

Reviewer #1: Yes

2. Has the statistical analysis been performed appropriately and rigorously? 

Reviewer #1: Yes

3. Have the authors made all data underlying the findings in their manuscript fully available?

Reviewer #1: Yes

4. Is the manuscript presented in an intelligible fashion and written in standard English?

Reviewer #1: Yes

5. Review Comments to the Author

Reviewer #1: Paradinas et al., present a simulation study to assess how random spatial effects represent missing co-variates using two spatial models: a Generalised Additive Model (GAM) and a Bayesian spatial model (INLA). They demonstrate that fitted spatial effects resemble the sum of the unaccounted covariate surfaces in each model.

The simulation approach is relatively simple (as acknowledged by the authors in the discussion) but clearly explained and relevant for the aims of the study. I have no major concerns with this study, however, I think the manuscript could benefit from some additional points in the introduction and discussion in order to further appeal to ecologists who may want to consider the results from this interesting study (see specific comments in attached file).

6. PLOS authors have the option to publish the peer review history of their article (what does this mean?). If published, this will include your full peer review and any attached files.

Reviewer #1: No

---

## [Author Response · Author response to Decision Letter 0]

23 Nov 2022

Review: Understanding spatial effects in species distribution models

Summary: 

Paradinas et al., present a simulation study to assess how random spatial effects represent missing co-variates using two spatial models: a Generalised Additive Model (GAM) and a Bayesian spatial model (INLA). They demonstrate that fitted spatial effects resemble the sum of the unaccounted covariate surfaces in each model.

The simulation approach is relatively simple (as acknowledged by the authors in the discussion) but clearly explained and relevant for the aims of the study. I have no major concerns with this study, however, I think the manuscript could benefit from some additional points in the introduction and discussion in order to further appeal to ecologists who may want to consider the results from this interesting study (see specific comments below).

Abstract & general comments to consider throughout the text

I’m not sure I agree with “most SDMs include spatial effects”. Consider softening with “SDMs can account for random spatial effects which may improve prediction… etc”. Note that the use of random spatial effects is sometimes used in linear modelling (Bayesian or not) but less so for machine learning which are popular SDM algorithms (e.g., Random Forest, Boosted Regression Trees, Maxent, etc) – you could consider including this point somewhere in the MS – possibly in the discussion?

We agree with the reviewer that it is better to soften the wording and we decided to use his recommendation. However, we may not comment anything on the no-use of spatial effects (as in geostatistics or 2D splines) in ML, maxent… methods as, in our opinion, does not provide any valuable point to the discussion topic.

Note in my suggested wording above, I use the term “random spatial effects”. If the authors agree with this clarification. Suggest this term is used throughout the text, or at first mention to make clear. 

We decided not to use the term “random”. Even though Bayesian spatial effects are structured random effects, not all spatial effects are random. For instance, are 2D splines random effects?

I’m assuming that throughout that the manuscript the authors are referring to SDMs fitted with occurrence data (the most common SDMs in my experience). If so, please make this very clear early on. If not, then further explanation is needed for how to interpret statements about Type I error for SDMs fitted with occurrence data AND with those fitted with abundance data (since for the latter, you may need to approach this as a hurdle model or delta model). 

The authors refer to “Type I errors”. I don’t disagree that this term is an accurate description, but I think many ecologists in the context of SDMs (where the null hypothesis is not clear) would have difficulty interpreting the statement. For example, if the authors are referring to SDM fitted with occurrence data, would type I refer to “false positive rate” and “false negative rate” i.e., misclassification of presences (predicted 0 when it should 1) and misclassification of absences (predicted 1 when it should 0)? Can you try and link this explanation to commonly used model fit metrics for SDMs such as TSS (True Skill Statistic) which specifically deal with sensitivity and specificity?

This is an interesting comment. In fact, this is an abundance SDM using a Poisson. By Type I error, we mean “mistaken rejection of an actually true null hypothesis”, or in SDM terms, identifying a environmental driver as significant while in reality isn’t. This should be the same whether the response variable is occurrence, abundance, or biomass, thus should be the same in a hurdle model.

Again, I’m not sure I agree with the statement “Ecologists tend to try ecologically interpret the spatial patterns displayed by the spatial effect” suggest softening it to “In some cases ecologists try to ecologically interpret the spatial patterns displayed by the spatial effect” since some SDM studies are principally concerned with predictive ability rather than interpretation of drivers.

We agree that softening our original terms is good. We have now changed these as suggested.

Suggest instead of “This study wants to provide a practical demonstration” the authors use “This study aims to provide a practical demonstration”

We thank the reviewer. It reads better now :)

Suggest that the authors generalise the description of the methods to “Generalised Additive Model” and shortened to “GAM” and “Bayesian spatial model” when generally referring to the methods in the intro / discussion – in my experience this terminology is more informative / commonly used than “2D smoothing splines” and “geostatistic model” respectively. 

Here, as with the random spatial effect suggestion, we may prefer to keep it as it is. Indeed, the term GAM is widely used to refer to models that include splines, but reality is that a gaussian linear model is also a GAM. Similarly, spatial model may refer to different things, as there are different spatial models (lattice, point, etc.). In fact, geostatistics do not necessarily have to be applied in the Bayesian framework. Therefore, we prefer to be specific about the methodology and use geostatistics and 2D splines.

Introduction:

L 8-10: I think the authors need to broaden this out to link this more with ecology. Suggest rejigging this sentence to reflect the below points

1. ideally co-variates (spatial and non-spatial) would fully explain the relationship between species occurrence or abundance with biotic and abiotic factors (you could briefly explain this in the context of ecological niches?). 

2. In practice this isn’t possible since it’s really complicated (you mention this in the discussion) and the co-variates we have aren’t always accurate. 

3. Random spatial or spatial/temporal effects can be incorporated in SDMs to proxy for unaccounted for processes which have a spatial, temporal, spatial-temporal structure (i.e., missing co-variates / unaccounted for interactions between co-variates). 

We thank the reviewer for this comment as we believe that it has helped improve the MS. The edited text is in the first paragraph of the MS and highlighted in blue

L8 – 9: Can you also provide examples of each co-variate in brackets? E.g., assuming that “non-spatial covariates” would be things like year, month?

Yes, indeed, this was a good suggestion, but we have decided to remove this sentence after expanding the text based on the previous suggestion. We believe that the text reads a lot better now, and the distinction between spatial and non-spatial covariates is not that important for the purpose of this MS.

L14 -16: my above point about proxying for unaccounted for co-variates is relevant here. I agree with the authors but suggest that you clarify that these are due to “unaccounted environmental or biotic drivers” which are often variables which are hard to measure or estimate, e.g., “geographical range dispersion process, e.g. colonisation and other highly dynamic processes such as wind and current”. 

We have now re-written this paragraph. The edited part is highlighted in blue.

L17 – 23: very good points about scale

L43: suggest remove “novel”

L46: suggest remove “rather”

L47: I’m not familiar with the terminology “model-based spatial models” Can you not simply call these “spatial models”? Or simplify some other way?

We have edited our text according to all the above suggestions. Model-based spatial model essentially refers to spatial models that include covariates as opposed to those spatial models that are purely spatial.

Simulation:

Suggest that you need to clarify somewhere in this section:

• whether your SDMs are occurrence

• that you calculated the residuals (table 2) and what these represent (i.e., deviations predicted from actual empirical values of data – which comes back to the units need to be clarified)

Our SDMs are Poisson models. This section begins mentioning “at each iteration we added a fixed number of new specimens…” which implies abundance, but we have now added Poisson spatial models in line 61. Regarding the mention about residuals, we made a mistake naming the column that way in tables 2, 3 and 4. Instead, we have now named it “dispersion” as it refers to the spatial aggregation derived from the simulated dispersion process of the species.

L50: depending on the formatting requirements of the journal, this may be better called “methods”

We agree with the reviewer and the section is now called “methods”.

L52: this is a good example of an often-omitted co-variate which random spatial effects are often interpreted as representing. Suggest you (briefly) revisit this in the discussion. 

Indeed. We have now mentioned this in the first sentence of the discussion section.

L64-66: see suggestions on model terminology. Also need to state whether you used default settings (which I assume you did). 

Yes indeed, this is now included in the MS

Results: 

L93 – 95: suggest this is discussion rather than results. 

Indeed, we have now moved this into the discussion

Discussion:

L99: suggest need a reference as an example for this statement 

It´s been hard to find a reference for this statement. However, it is a rather well

L104 -105: suggest, “Here, we did not attempt to account for all possible cases, instead, we aimed to illustrate our point using a simple and intuitive approach.”

We thank the reviewer for this suggestion. The entire sentence has been edited as suggested. 

L124 – 126: Agree that this can increase predictive capacity, but only within sampled areas (i.e., interpolation between sample points). Although extrapolation isn’t encouraged, in many cases SDMs are spatially predicted into areas with low samples where random spatial effects provide no additional benefit (especially if they override the effects of other co-variates which are important drivers). I think you could mention this caveat. That is, to use spatial effects with care when the aim if prediction. 

This is an interesting point, but we have decided not to discuss much about this in the MS. We did a bunch of trials but we finally concluded that they deviated the discussion, and therefore we have simply edited the text so that it is clear that spatial effects improve prediction within the sampled area.

---

## [Editor Report · Decision Letter 1]

1 Dec 2022

PONE-D-22-21589R1Understanding spatial effects in species distribution modelsPLOS ONE

Dear Dr. Paradinas,

Thank you for submitting your manuscript to PLOS ONE.  While you have largely met the reviewer's suggestions, in re-reading the manuscript I did pick up one further point.  IN the introduction you mention " one can accommodate the spatial or spatio-temporal structure of the data that is unaccounted by our covariates".  Post that the focus is on spatial effects.  Could you either remove the spatio-temporal comment or actually include how temporal changes to spatial structures (or real species- environment relationships) may alter your results.

We look forward to receiving your revised manuscript.

Kind regards,

Judi Hewitt

Academic Editor

PLOS ONE
---

## [Author Response · Author response to Decision Letter 1]

18 Apr 2023

This version contains the last edit requested by the editor. We have removed the mention of spatiotemporal effects as we finally decided that it would only confuse the reader.

---

## [Editor Report · Decision Letter 2]

25 Apr 2023

Understanding spatial effects in species distribution models

PONE-D-22-21589R2

Dear Dr. Paradinas,

We’re pleased to inform you that your manuscript has been judged scientifically suitable for publication and will be formally accepted for publication once it meets all outstanding technical requirements.

Kind regards,

Judi Hewitt

Academic Editor

PLOS ONE
---

## [Editor Report · Acceptance letter]

19 May 2023

PONE-D-22-21589R2 

Understanding spatial effects in species distribution models 

Dear Dr. Paradinas:

I'm pleased to inform you that your manuscript has been deemed suitable for publication in PLOS ONE. Congratulations! Your manuscript is now with our production department. 

Kind regards, 

on behalf of

Dr. Judi Hewitt 

Academic Editor

PLOS ONE